# Quantum dynamics simulation of exciton-polariton transport

Niclas Krupp [1] ✉, Gerrit Groenhof [2] & Oriol Vendrell [1] ✉

Strong coupling between excitons and confined modes of light presents a promising pathway to tunable and enhanced energy transport in organic materials. By forming hybrid light-matter quasiparticles, exciton-polaritons, electronic excitations can traverse long distances at high velocities through ballistic flow. However, transport behavior of exciton-polaritons varies strongly across experiments, spanning both diffusive and ballistic transport regimes. Which properties of the material and light-modes govern the transport behavior of polaritons remains an open question. Through full-quantum dynamical simulations we reveal a strong dependence of polariton transport on vibronic interactions and static disorder within molecules in both ideal and lossy cavities. Specifically, we show that intramolecular vibrations mediate relaxation processes that alter polariton composition, lifetime and velocity on ultrafast timescales. Analysis of the propagating wavepacket in position and momentum space provides mechanistic insight into the robustness of ballistic flow of exciton-polaritons found experimentally under cryogenic conditions.

Since the seminal observation of Rabi splitting in a semiconductor microcavity in 1992[1], the field of exciton-polaritons (EPs)−hybrid light-matter states involving electronic material excitations and quantized modes of light−has been rapidly evolving in both theory and experiment. Nowadays, experimental platforms which typically consist of a Fabry-Pérot (FP) microcavity filled with inorganic or organic emitters can routinely achieve strong coupling between material and photonic subsystems[2–4]. Consequently, a growing body of research is now dedicated to the characterization and application of the unique properties of EPs, bridging condensed matter physics, material science, and chemistry. EPs have low effective mass and exhibit adjustable dispersion relations that differ from those of the bare material. They can also be selectively optically excited, which enables the development of highly tunable platforms for exciton condensation[5–11] and transport processes[12–17]. Hybridization of excitons with cavity photons might overcome notoriously inefficient and diffusive exciton transport in organic materials, achieving ballistic energy flow in polaritonic devices[18]. Thus, polariton transport presents a strategy to improve and steer exciton mobility in semiconductors, which is of high relevance for organic photovoltaics[19]. Indeed, experimental studies on EP transport could show propagation of EPs over long distances, and over time spans which exceed the bare cavity photon lifetime[12–15].

However, the mechanism and efficiency of polariton transport are still unclear: ballistic EP flow has been observed in inorganic semiconductor polaritons at cryogenic temperatures[15,20], and for organic Bloch surface wave polaritons[14]. In these cases, polaritons move at the expected group velocity obtained from the gradient of the dispersion curve. In contrast, other studies in organic J-aggregate microcavities measured diffusive EP transport at reduced velocities below the expected group velocity[16]. Current state-of-the-art ultrafast microscopy experiments on inorganic microcavity polaritons and organic Bloch surface wave polaritons found a transition from ballistic to diffusive transport, which can be tuned via the exciton content of EP[12,13]. Here, ballistic transport at high group velocities is maintained for up to 50% exciton content, while more excitonic EPs propagate diffusively at lower-than-expected velocities. In the low-temperature regime, steady-state measurements[15,20] and recent time-resolved experiments[12] both have found ballistic flow and no group velocity reduction, even at exciton contents as high as 88%[12].

Organic condensed-phase microcavities exhibit static and dynamic disorder. Static disorder, on the one hand, is connected with

[1]Theoretische Chemie, Physikalisch-Chemisches Institut, Universität Heidelberg, Heidelberg, Germany. [2]Nanoscience Center and Department of Chemistry, University of Jyväskylä, Jyväskylä, Finland. ✉e-mail: niclas.krupp@pci.uni-heidelberg.de; oriol.vendrell@uni-heidelberg.de

the material-dependent inhomogeneous broadening of the excitonic resonance. Dynamic disorder, on the other hand, results from molecular nuclear dynamics and causes time-dependent fluctuations in electronic transition energies and cavity-molecule interaction. Nuclear dynamics can either occur thermally-activated in the electronic ground state, or in the excited states due to different topologies along ground and excited state potential energy surfaces at the Franck-Condon point, i.e., due to vibronic coupling. Both types of disorder lead to elastic and inelastic scattering processes of EPs, such as vibration-assisted scattering (VAS) and radiative pumping (RP), which are ubiquitous in semiconductor cavity systems[21–25]. Yet, their impact on EP dynamics is still debated in the field. Focusing on EP transport, dynamic and static disorder effects—in particular EP-phonon and EP-defect scattering—are speculated to play a role in determining the ballistic to diffusive crossover, group velocity renormalization, and the feeding of population from the exciton reservoir back to the coherent polaritonic manifold[12,17,26,27].

From a theory perspective, studies based on semiclassical Ehrenfest-type dynamics[17,28,29] could gain first insights into effects of cavity losses and reversible vibration-driven population transfer to dark states on EP transport. However, simulations entirely based on quantum mechanics and including molecular vibrations and dispersive cavity modes have not been attempted yet, but are clearly needed for a thorough mechanistic understanding of polaritonic transport processes. Here, we report full-quantum dynamical simulations of exciton transport in molecular arrays coupled to perfect and lossy FP cavities. Employing the highly compact multi-layer multiconfiguration time-dependent Hartree (ML-MCTDH) wavefunction ansatz[30–33] enables us to account for the multi-mode nature of the cavity, and more than 200 molecules with vibrational degree of freedom. The molecular ensemble-cavity wavefunction is then propagated according to the ML-MCTDH equations of motion, which are obtained from the time-dependent variational principle[30]. With this efficient and accurate simulation protocol at hand, we can closely examine the impact of dynamic and static disorder, radiative decay, and laser excitation parameters on EP transport in an organic microcavity within a fully quantum-dynamical framework.

Experiments under cryogenic conditions are closely comparable to our quantum-dynamics simulations at 0 K. Our results thus offer detailed insight into preserved ballistic transport of polaritons in the low-temperature regime. Importantly, our full quantum-dynamical simulations support the existence of ballistic transport even for highly excitonic initial states, while uncovering the role of vibronic coupling and static disorder at low temperatures. Molecular vibrations which couple to electronic transitions, can improve EP transport by mediating relaxation of highly excitonic wavepacket components toward increasingly photonic states with higher group velocity. By considering experimentally relevant excitation schemes with explicit laser pulses, we identify regimes in which this vibronic-enhancement mechanism is operative. Moreover, we find that this effect can revert and lead to a slow-down of EP transport, depending on the targeted polaritonic branch. Static energetic disorder, on the other hand, results in fast localization of the propagating polariton, which leads to a transition from a ballistic to diffusive transport regime. Nonetheless, vibronic-enhancement stays in competition with the localization effect such that the ballistic regime can be extended for polaritons with exciton contents above 50%. Finally, our simulations suggest that resonant excitation of EPs on the LPB close to 50% exciton content, provides a favorable trade-off between high propagation velocities and prolonged polariton lifetime in lossy cavities with realistic radiative lifetimes.

## Results

First, we study the quantum dynamics of EP transport an "ideal" organic microcavity model without static disorder (cf. Fig. 1a). The cavity and molecular parameters are chosen to closely resemble the Rhodamine microcavity model of ref. 17, albeit with a simpler vibrational structure: in each molecule, a vibrational mode couples to the electronic transition which is parameterized by the vibronic coupling constant $\kappa$. Details of the model are given in "Methods".

Due to their hybrid light-matter nature, EPs can propagate along the in-plane direction of the cavity with group velocity $v_{gr}^{LP/UP} = \partial \omega_{LP/UP} / \partial k_x$, where $\omega_{LP/UP}$ denotes the dispersion relation of the upper polariton branch (UPB) and lower polariton branch (LPB). The polariton dispersion, group velocity, and exciton content in the absence of vibronic coupling are shown in Fig. 1c–e, and play a vital role in our analysis.

Experimentally, EP transport is initiated either via off-resonant[13,15,16] or resonant excitation[12,14,20,34]. In the first case, a focused laser pulse pumps the transition to a high-lying electronic excited state from where the system relaxes incoherently to the $S_1$ state. In our simulations, off-resonant excitation is modeled by instantly promoting a single molecule to the $S_1$ state while all other constituents of the system remain in their respective ground state[17]. In the second case, incidence angle and frequency of the laser pulse target a specific point on the polaritonic dispersion curve, coherently exciting a polaritonic wavepacket centered around a well-defined in-plane momentum $k_x^{(0)}$ and energy $\hbar\omega_L$. All relevant model and laser parameters are summarized in Table 1.

After excitation, we trace the spatiotemporal polariton dynamics by computing the real-time and real-space resolved polaritonic density $|\psi_{pol}(x_j, t)|^2$ and the individual contributions from the photonic ($|\psi_{phot}(x_j, t)|^2$) and molecular ($|\psi_{mol}(x_j, t)|^2$) subsystems, as well as the momentum-resolved photonic density $|\psi_{phot}(k_{x,p}, t)|^2$. These quantities can be readily leveraged from the propagated ML-MCTDH wavefunction by projecting onto localized photonic and molecular excitations, or onto photonic momentum-eigenstates[17].

### Off-resonant excitation

Sudden localized excitation of a single molecule in real-space creates a delocalized excitation in energy-momentum space, affording a superposition of a broad range of LPB and UPB polaritons. Consequently, components with a broad range of group velocities propagate away from the initial off-resonant excitation spot at $x_0 = 2.5$ μm.

Without vibronic coupling ($\kappa = 0$), (Fig. 2a, upper panel), a wavefront moving linearly at the maximum LP group velocity $v_{gr}^{LP, max} = 67.9$ μm ps$^{-1}$ is visible in the polaritonic as well as photonic and molecular densities, while zooming in on the photonic real-space density reveals the presence of an additional—but faint—wavefront at the maximum UP group velocity which is close to the speed of light ($v_{gr}^{UP, max} = 235$ μm ps$^{-1}$).

As such, when vibrations do not couple to the electronic transition, transport after off-resonant excitation is driven mainly by the photonic part of the wavefunction: in Fig. 2a, c mean position and mean square displacement (MSD) of the photonic subsystem exceed those of the molecular subsystem significantly, and a large part of the molecular density remains at the initial excitation spot.

This can be explained by the fact that the initial single-molecule excited state has largest overlap with polaritonic states with high molecular contribution, which are located on the far right of the LPB dispersion, and the far left of the UPB dispersion curve (cf. Fig. 1d). Thus, the initial molecular excitation mostly spans polaritonic states with vanishing group velocity, as can be seen from Fig. 1e. Fast moving components close to the speed of light originate from high in-plane momentum UPB states which have negligible excitonic contribution (Fig. 1d, e). Therefore, their signatures are weak and only visible in the photonic density.

The presence of dynamic disorder due to non-zero vibronic coupling $\kappa$ drastically changes the polariton transport dynamics: increasing $\kappa$ in Fig. 2a shifts formerly stationary or slowly propagating

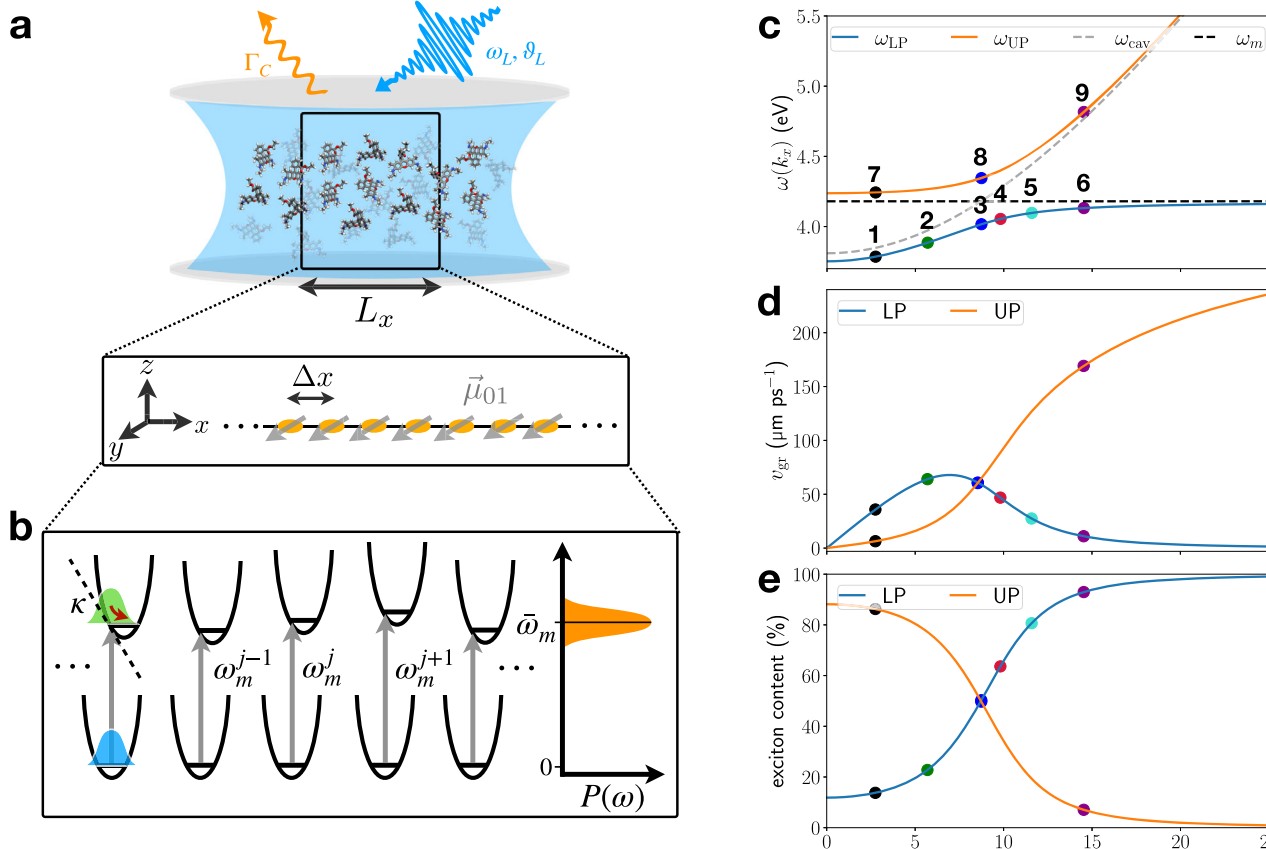

**Fig. 1 | Overview of exciton-polariton transport model. a** Schematic of an organic microcavity filled with an ordered molecular ensemble in which all molecular TDMs $\vec{\mu}_{01}$ are aligned with the cavity polarization direction. **b** Static and dynamic disorder are included in the model by sampling the molecular electronic excitation energies $\omega_m^j$ from a Gaussian distribution $P(\omega)$, and by introducing vibronic coupling (characterized by $\kappa$), respectively. The latter results in nuclear motion in the excited electronic state. **c** Polariton dispersion curve $\omega_{\mathrm{LP/UP}}(k_x)$ with bare molecular $\omega_m$ and bare cavity $\omega_{\mathrm{cav}}$ dispersion. **d** Corresponding exciton content and **e** group velocities of lower and upper polariton branches. Colored circles in (**c–e**) mark the polaritonic states targeted through resonant excitation by an external laser pulse.

components toward the wavefront, which moves at $v_{\mathrm{gr}}^{\mathrm{LP,\,max}}$. This enhances the mobility substantially, as indicated by the mean polaritonic position and MSD in Fig. 2a, c. Furthermore, the mobility-enhancement occurs dynamically, i.e., the mean group velocity increases over time, which can be seen from the increasing slope of time-dependent mean position $\langle x_{\mathrm{pol}}\rangle(t)$ in Fig. 2a. The enhancement is more pronounced in the material subsystem, pointing at a vibronically-mediated mechanism that dynamically increases the mobility of exciton-like polaritons.

This mechanism demands closer analysis. Inspecting the total population of cavity modes and molecular excitonic states in Fig. 2b reveals an increasing photonic content over time for non-zero $\kappa$. For the largest vibronic coupling strength ($\kappa = 80$ meV), the light-matter composition changes from purely excitonic at the beginning of the simulation to a roughly 60/40 photonic/excitonic composition after approximately 100 fs. Moreover, the mean in-plane momentum of cavity modes in Fig. 2d stagnates around 8 μm$^{-1}$ for $\kappa = 0$ meV but decreases noticeably to approximately 6 μm$^{-1}$ for $\kappa = 80$ meV. These findings suggest vibration-driven relaxation on the LPB as the underlying mechanism for the mobility-enhancement, whereby population from highly excitonic, near-stationary states is transferred toward more photonic states with large group velocity (cf. Fig. 1d).

For $\kappa \neq 0$, the initial wavepacket formed by off-resonant excitation, no longer corresponds to a superposition of excitonic-photonic eigenstates. Instead, intramolecular vibrations couple states on UPB and LPB, inducing VAS and RP processes. VAS and RP have been amply discussed for organic microcavity EPS[21–23,26,35,36], and are relevant for polariton condensation[5,37]. Highly excitonic polaritonic states with large in-plane momentum on the LPB can scatter into a lower-momentum region with increased photonic character and increased group velocity, resulting in a dynamically increasing mean velocity as indicated by the change in the slope of the red line in Fig. 2a. Scattering processes on the UPB, on the other hand, do marginally contribute to this transport-enhancement since fast-propagating, increasingly photonic states are located at high in-plane momenta and possess high energy, making them hardly accessible from the off-resonantly excited initial state. Apart from that, scattering from the UPB to the LPB can contribute, as we discuss in the next section.

For better understanding of the underlying relaxation process, we inspect the momentum-resolved photonic populations $|\psi_{\mathrm{phot}}(k_x)|^2$ at $t_i = 5$ fs and $t_f = 240$ fs for various detunings $\Delta_{\mathrm{exc}}$ between the excitonic resonance and the fundamental cavity frequency $\omega_0 = 3.81$ eV. This tunes the in-plane momentum $k_x^{\mathrm{res}}$ at which cavity mode and exciton are resonant, from low to high in-plane momentum, cf. Fig. 2e. We find in all cases a clear decrease in population of momentum-states to the right of the resonant in-plane momentum, and an increase to the left. Accordingly, mainly states with $k_x \leq k_x^{\mathrm{res}}$, occasionally referred to as "bottleneck" polaritons[15,23], participate in the vibration-driven relaxation. Moreover, these states do not relax down to $k_x = 0$ but rather accumulate in the intermediate region $0 < k_x < k_x^{\mathrm{res}}$ of the LPB where molecular contributions are still appreciable and group velocities are high (cf. Fig. 1d, e).

**Table 1 | Cavity ($\omega_O$, $n$, $\tau_{cav}$, $\hbar\Omega_R$), molecular ($\omega_m$, $\bar{\omega}_m$, $\sigma_m$, $\omega_{vib}$, $\kappa$, $\Delta x$), and laser parameters ($F_t$, $F_x$, $t_O$, $x_O$) relevant to simulations**

| Parameter | Value |
|---|---|
| $\omega_O$ | 3.81 eV |
| $n$ | 1.0 |
| $\hbar\Omega_R$ | 328 meV |
| $\tau_{cav}$ | 24 fs |
| $\omega_m = \bar{\omega}_m$ | 4.18 eV |
| $\omega_{vib}$ | 74 meV |
| $\sigma_m$ | 30 meV |
| $\kappa$ | 0 meV, 50 meV, 80 meV |
| $\Delta x$ | 250 nm |
| $F_t$ | 18 fs |
| $F_x$ | 625 nm |
| $t_O$ | 40 fs |
| $x_O$ | 2.5 μm |

$F_t$ and $F_x$ denote the temporal and spatial full width at half maximum (FWHM) of the Gaussian laser pulse, respectively. The maxima of temporal and spatial Gaussian envelopes are found at $t_O$ and $x_O$, respectively.

Further relaxation is hindered by the decreasing efficiency of exciton-mediated scattering when the photonic content of involved states becomes large[11,23,35]. This effect is known as the "phonon bottleneck" in the context of EP condensation[5,18]. Note that the wavevector-resolved photonic population cannot capture highly excitonic polaritons due to their vanishing photonic contribution. Nonetheless, the disappearing stationary component at $x_0$ of $|\psi_{mol}(x, t)|^2$ for $\kappa = 50$ meV and $\kappa = 80$ meV in Fig. 2a indicates that they undergo relaxation on the LPB as well.

**Resonant excitation**
In the following, we investigate the relevance of molecular vibronic coupling in a wide range of resonant excitation scenarios targeting LPB and UPB. To this end, eight laser pulses are considered with pulse parameters (except photon energy and incidence angle, which are varied) given in Table 1. The resonantly targeted points in the [$\omega(k_x)$, $k_x$] plane are shown in Fig. 1c. First, we discuss two limiting cases on each branch: a highly photonic and a highly excitonic wavepacket, which, on the LPB, are found at small and large wavevectors, respectively. The situation inverts on the UPB, where the photonic content increases towards large wavevectors.

The LPB dispersion relation at small wavevectors is parabolic, such that coherently-excited wavepackets in this region display ballistic transport and dispersive behavior[38]. Our full-quantum dynamical results reproduce this for resonant excitation targeted at $k_x^{(0)} = 2.75$ μm$^{-1}$ (pulse 1) on the LPB, corresponding to an exciton content of 13.8% (cf. Fig. 1d). In Fig. 3a the mean position of the polaritonic density moves linearly, corresponding to quadratic growth of the MSD, while the width of the wavepacket spreads visibly during propagation in a lossless cavity. Polaritonic densities in both real and momentum space are identical for $\kappa = 0$ meV and $\kappa = 80$ meV (Fig. 3a, c); the wavevector-resolved cavity-mode population stays constant after the pulse, showing no signatures of relaxation (Fig. 3c). As discussed for the off-resonant case, the efficiency of vibration-mediated scattering processes decreases significantly towards the small-wavevector region on the LPB due to increasing photonic content. Consequently, the ballistic dispersive transport is unaffected by vibronic coupling in this regime.

Strong differences are found when the laser pulse addresses the highly excitonic region on the LPB, exciting at $k_x^{(0)} = 14.5$ μm$^{-1}$ (pulse 6) corresponding to an exciton content of ~90%. For $\kappa = 0$ meV transport

proceeds ballistically, Fig. 3a, at low group velocity, but dispersion during propagation is less pronounced due to the non-parabolic, flat LPB dispersion relation in the large in-plane momentum region. Vibronic coupling significantly changes this behavior. Again, strong vibronic transport-enhancement becomes apparent from the polaritonic real-space density and mean position in Fig. 3a. Wavevector-resolved cavity populations in Fig. 3c reveals rapid relaxation of high in-plane momentum components of the LPB wavepacket to faster-propagating polaritons close to the "bottleneck region" below $k_x^{res} = 8.64$ μm$^{-1}$ as the origin of the enhancement. These findings verify the participation of highly-excitonic LPB states in the vibration-mediated relaxation as discussed for off-resonantly excited polaritons.

Targeting the corresponding UPB states with pulses 7 and 9 results in an inverted situation: Now, excitation at large wavevectors affords a highly photonic EP wavepacket with large group velocity which is shielded from molecular vibronic effects through its low exciton content of ~7%. Consequently, we find ballistic transport independent of the vibronic coupling strength in Fig. 3b. In contrast, low-wavevector (pulse 7) excitation creates a predominantly excitonic EP wavepacket with low group velocity. Again, vibronic coupling results in relaxation of the wavepacket on the UPB to the "bottleneck region" in Fig. 3d which follows a similar fate as the LPB wavepacket excited by pulse 6 in Fig. 3c. As a result, the propagation velocity is significantly enhanced, producing very similar trajectories for initially highly excitonic wavepackets on the LPB and UPB (pulse 6 in Fig. 3a and pulse 9 in Fig. 3b). This suggests that vibronic interactions within molecules can efficiently funnel population towards the high-mobility region on the LPB close to $k_x^{res}$ from both UP and LP states.

Next, we target the high-mobility region directly by excitation to the UPB and LPB at the resonant wavevector $k_x^{res}$ (pulses 3 and 8 in Fig. 1c). Although possessing the same nominal group velocity and equal light-matter composition in the absence of vibronic coupling, UP and LP states at $k_x^{res}$ undergo distinct relaxation dynamics, both resulting in a slow-down of polariton propagation when vibronic coupling is included, as seen in Fig. 4c. This is in stark contrast to the previously discussed highly excitonic UP and LP states which both follow similar relaxation pathways and exhibit vibronically-enhanced transport.

After UP excitation, vibronic coupling leads to the appearance of non-propagating "vertical" features which lack any contribution from the cavity modes (Fig. 4b). Still, a component propagating at the expected group velocity persists in the photonic and polaritonic real-space densities. This results in the onset of an apparent diffusive transport, as indicated by the linear growth of the polaritonic and molecular MSDs in Fig. 4c toward the end of the simulation.

We can attribute this apparent slow-down effect for UP excitation to non-radiative decay from the UPB towards the dark states driven by vibronic motion. Polariton dynamics around the resonant wavevector can be regarded in close analogy to that of many molecules electronically coupled to a single resonant cavity mode. In that case, vibronic interactions in molecules have been found to support fast and efficient decay to the dark-state manifold from the UP state[39,40].

For LP excitation at $k_x^{res}$, such vertical features are not visible in the polaritonic density (cf. Fig. 4a). Instead, fractions with lower group velocities depart from propagating wavefront over time, indicating population transfer to polaritonic states with lower group velocities. Accordingly, intraband relaxation on the LPB dominates the dynamics after LP excitation, which can be traced through the wavevector-resolved photonic density in Fig. 4d. While this mechanism generally enhances the transport as observed in this study for off-resonant and resonant excitation schemes, it can only access slower-propagating states when starting at the resonant wavevector, i.e., close to the maximum of the LPB group velocity (cf. Fig. 1e).

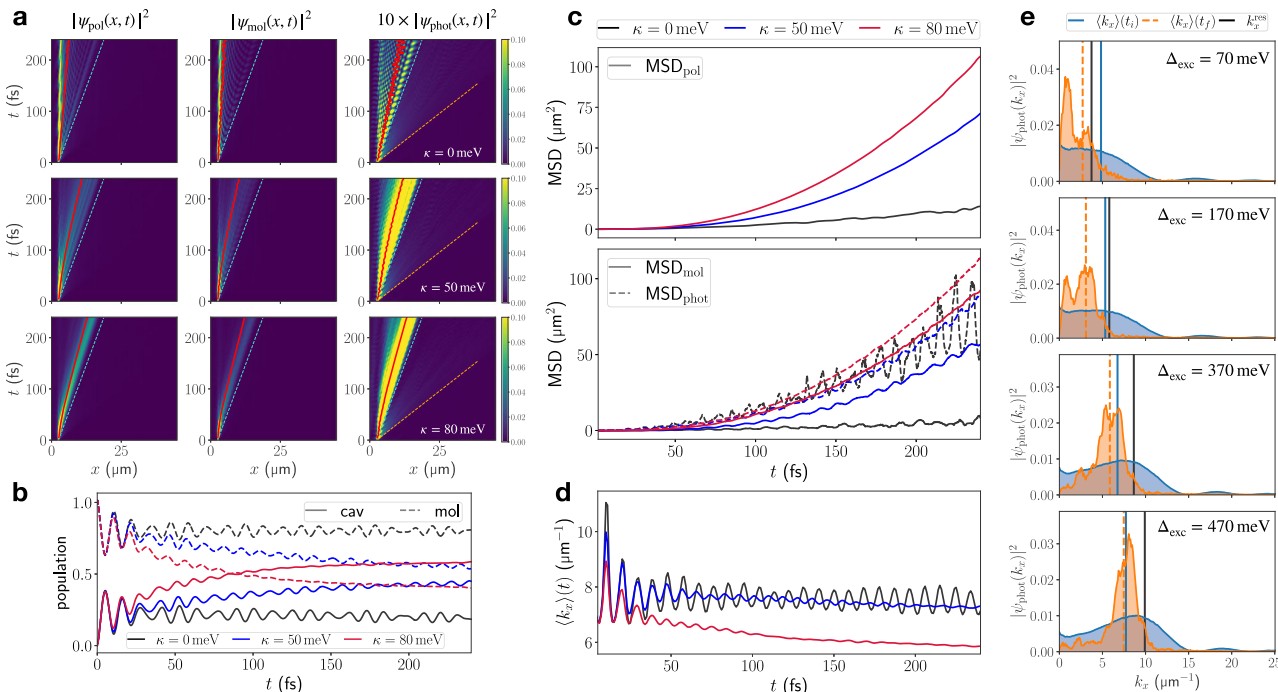

**Fig. 2 | Impact of vibronic coupling on EP transport after off-resonant excitation. a** Real-space resolved total polaritonic density $|\psi_{pol}(x,t)|^2 = |\psi_{mol}(x,t)|^2 + |\psi_{phot}(x,t)|^2$, molecular and photonic contributions $|\psi_{mol}(x,t)|^2$ and $|\psi_{phot}(x,t)|^2$, respectively, for vibronic coupling strengths $\kappa = 0, 50, 80$ meV. Cyan and orange dashed lines indicate linear motion at the maximum group velocities of LPB (67.9 μm ps$^{-1}$) and UPB (235 μm ps$^{-1}$), respectively. Red lines trace the expectation value of the position operator. **b** Time-dependent population of excited cavity modes ("cav") and molecular excitons ("mol"). **c** MSDs extracted from real-space resolved densities in (**a**). **d** Expectation value of cavity mode in-plane momentum for propagating EPs in (**a**). **e** Initial and final momentum distribution for $\kappa = 80$ meV and various detuning $\Delta_{exc}$ of molecular electronic excitation energy with respect to cavity photon energy $\omega_0$. The corresponding resonant wavevector $k_x^{res}$ is indicated by a black vertical line. All distributions in (**e**) have been normalized.

Summarizing, our fully quantum-dynamical simulations of EP dynamics in a lossless cavity show that vibronic motion can strongly impact polariton transport depending on the initially targeted state. Vibronic interactions provide relaxation channels along and among polaritonic branches and dark states for appreciable exciton content ($\geq 20\%$) of the targeted EP state. Generally, fast-propagating states close to the polariton bottleneck can be accessed efficiently from exciton-like UPB and LPB states. This results in a vibronic-enhancement of EP transport. However, close to the inflection point of LPB and UPB trapping within the dark-state manifold and low group velocity EP states close to $k_x = 0$ become possible, resulting in a slow-down of EP transport.

## Impact of static disorder

Organic semiconductor materials typically display a high degree of structural and energetic disorder, resulting in inhomogeneous broadening of the excitonic resonance. Static disorder in general is known to have substantial impact on a variety of transport processes in organic materials, including charge and exciton transport in organic semiconductors as well as polariton transport[41–44]. Within our fully quantum-dynamical framework, we investigate how static disorder in conjunction with vibronic interactions, i.e., dynamical disorder, governs the behavior of EP transport.

We model static energetic disorder in an organic microcavity by considering a Gaussian distribution of molecular electronic excitation energies with standard deviation $\sigma_m = 30$ meV and mean value $\bar{\omega}_m = 4.18$ eV (cf. Fig. 1b). The corresponding linewidth (FHWM) of the excitonic resonance of $\approx 70$ meV is representative of organic materials used in strong-coupling experiments[45,46]. The results presented in Fig. 5 are averages over simulations performed with five disorder realizations.

In the presence of static disorder, the assumption of unidirectional transport can break down due to backscattering. In ref. 47 cavity modes with opposite in-plane momentum were found to participate strongly in the transport dynamics, and their inclusion was essential in accurately simulating EP mobility. We therefore modify the multimode cavity model to include negative in-plane momentum modes by choosing discretized $k_{x,p} = 2\pi p/L_x$ with $p \in \left[-\frac{M}{2}, \frac{M}{2} - 1\right]$ and keeping the number of modes equal. The wavevectors of this bidirectional cavity model extend from approximately $-12.5$ to $12.5$ μm$^{-1}$.

Indeed, backscattering of the propagating polariton is a dominant process in all simulations presented in Fig. 5. This is clearly indicated by the polariton wavepacket spreading to the left and right in real-space (Fig. 5a, d), and the population of negative-momentum cavity modes within 20–50 fs after resonant excitation to positive-momentum region of the LPB (Fig. 5b, e). Over the course of the propagation, the photonic momentum-space densities evolve towards an approximately symmetric distribution (with respect to $k_x = 0$), such that the average in-plane momentum $\langle k_x \rangle(t)$ tends to 0. As a result, the mean polariton position $\langle x_{pol} \rangle(t)$ approaches stationary behavior for longer propagation times (best visible in Fig. 5d), although a polaritonic wavepacket with positive $k_x$ is initially excited.

The (back)scattering processes induced by static energetic disorder within the molecules have a detrimental impact on polariton transport. Comparing the polaritonic MSDs for simulations with and without static disorder indicates suppressed transport in the disordered microcavity independent of the vibronic coupling strength (Fig. 5c, f). As discussed in refs. 22,44, inhomogeneous broadening of the exciton resonance increases scattering between the polariton and molecular excitations, thereby restricting the range of coherently propagating states on the LPB and increasing the number of incoherent, strongly localized states. This behavior, akin to Anderson

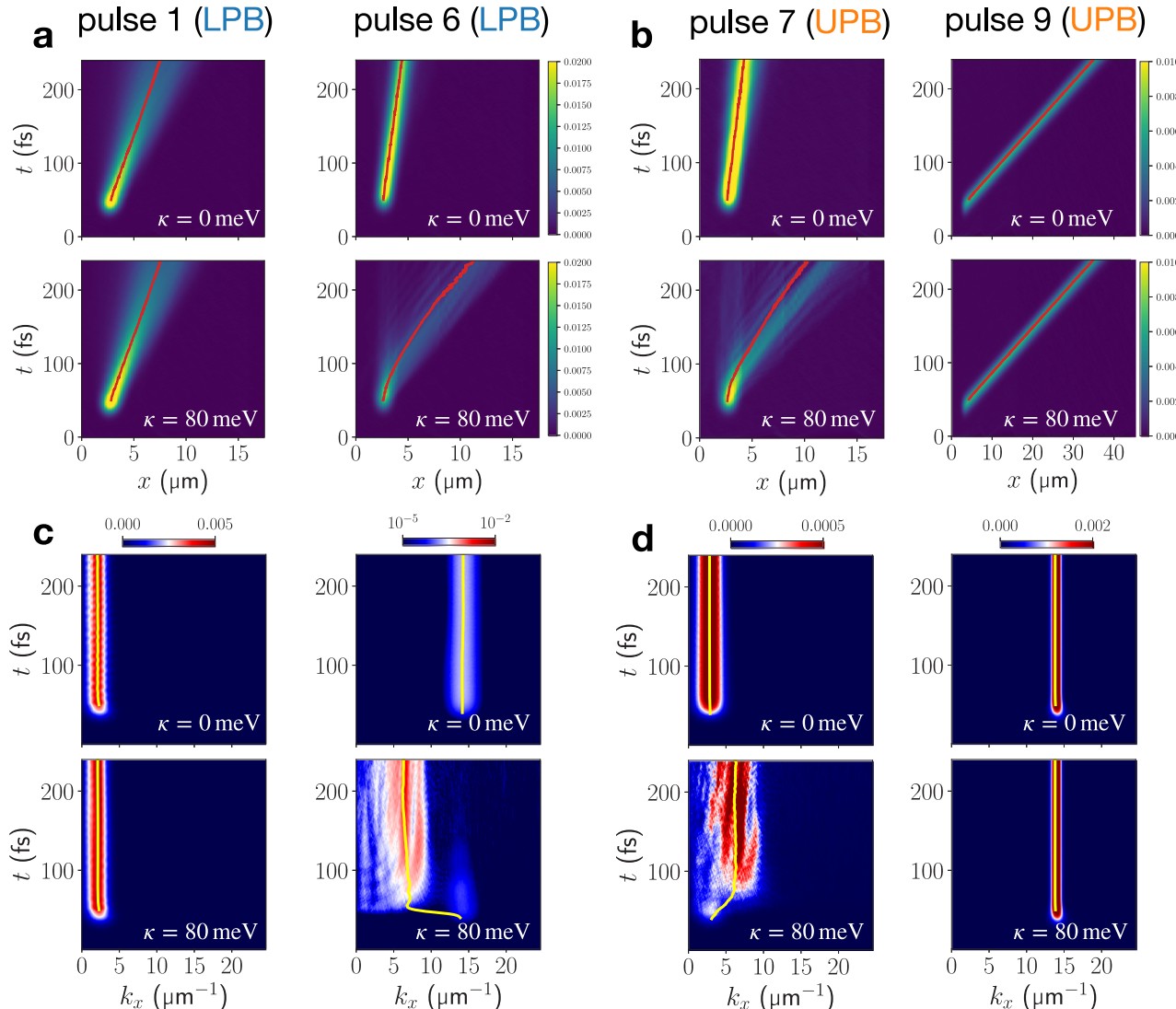

**Fig. 3 | Transport enhancement through vibronically-mediated relaxation of resonantly excited EP wavepackets. a** Real-space resolved total polaritonic density $|\psi_{\mathrm{pol}}(x, t)|^2$ for resonant-excitation to the LPB by pulse 1 and 6, at vibronic coupling strengths $\kappa = 0$ meV and $\kappa = 80$ meV. **b** Same as (**a**) but targeting the UPB by pulses 7 and 9. Red lines indicate the expectation value of the position operator. The momentum-resolved cavity mode populations corresponding to (**a**) and (**b**) are shown in (**c**) and (**d**), respectively. Yellow lines indicate the expectation value of the cavity-mode momentum operator.

localization[41], is visible in Fig. 5a, d where stationary population builds up next to the propagating wavefront.

Without vibronic coupling, this leads to the onset of diffusive transport after a short ballistic phase, ca. 50 fs after the pulse as indicated by the change from quadratic to linear growth of the MSDs in Fig. 5c, f at 100–125 fs. Localization is more pronounced for pulse 4 since the impact of molecular disorder is increased due to the higher exciton content. In that case, polariton transport is most suppressed (Fig. 5f). These findings are in excellent agreement with previous studies on polariton transport in statically disordered systems in the absence of vibronic coupling[47–49]. Note that the previously proposed (static) disorder-enhanced transport regime[44,49,50] is not observed here, since its onset occurs when the inhomogeneous broadening is comparable to the Rabi splitting[47,50] whereas here $\sigma_m / \Omega_R \approx 10\%$.

When vibronic interactions are included ($\kappa = 80$ meV), transport behavior differs strongly for pulses 3 and 4. In the subsection "Resonant Excitation", vibronic coupling has been found to result in a slow-down of EP transport after exciting to the LPB at the resonant wave-vector with pulse 3. Vibronic effects hence amplify the static disorder

effect for this excitation. Therefore, diffusive transport at a strongly reduced rate is observed in Fig. 5c.

In contrast, when targeting polaritons on the LPB with larger in-plane momenta, vibronic coupling has the opposite effect, enhancing transport by connecting slowly propagating exciton-like polaritons with the photon-like high-mobility region on the LPB (cf. pulse 6 in Fig. 3). This effect persists in the presence of static energetic disorder, as indicated by the emergent population of cavity modes in the bottleneck region around $\pm 6\ \mu\mathrm{m}^{-1}$ in Fig. 5e at 100 fs. Vibronic relaxation in combination with polariton backscattering results in a rapid "symmetrization" of the momentum-space populations, affording a relatively broad distribution with near-vanishing mean-momentum within ~50 fs after excitation.

Interestingly, this leads to a prolonged ballistic expansion phase with a larger final MSD compared to the disordered model without vibronic coupling (Fig. 5f): vibronic population transfer on the LPB accesses states which not only possess a higher intrinsic group velocity but are also less prone to scattering due to their increased photonic character. The molecular and photonic contributions to the MSD

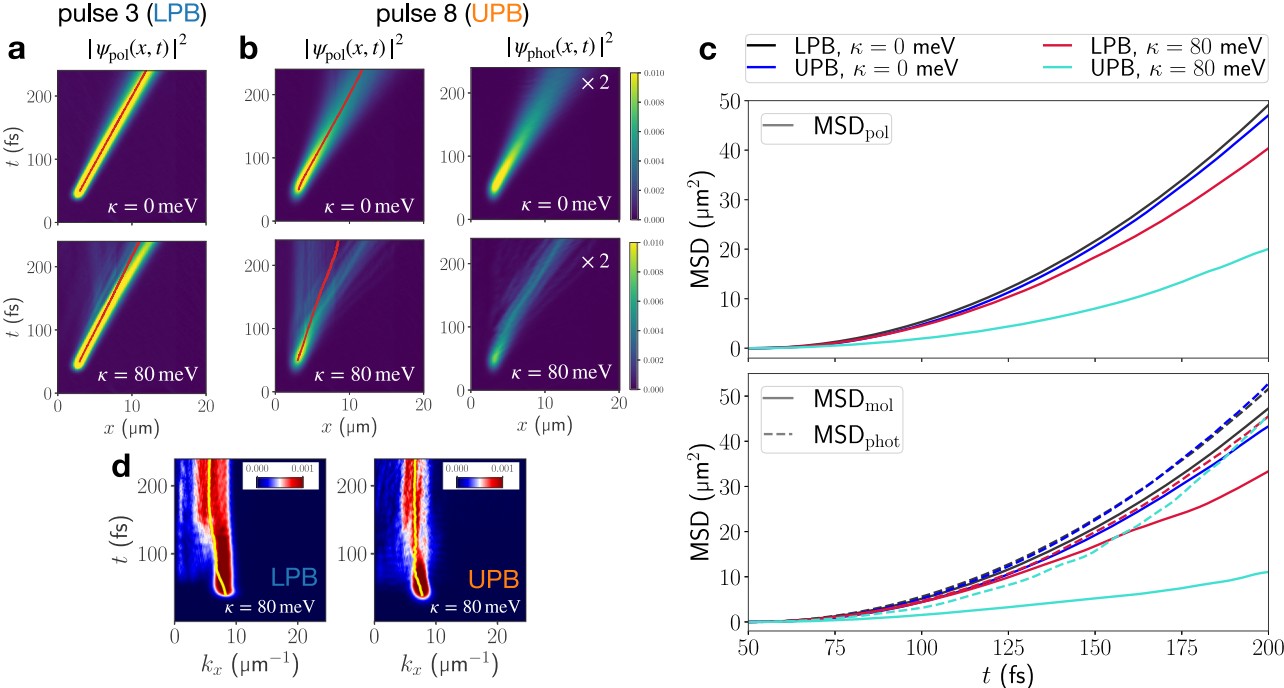

**Fig. 4 | EP propagation ensuing excitation at resonant wavevector. a** Real-space resolved total polaritonic density $|\psi_{\mathrm{pol}}(x, t)|^2$ for resonant-wavevector excitation to the LPB. **b** Real-space resolved total polaritonic density $|\psi_{\mathrm{pol}}(x, t)|^2$ and photonic contribution $|\psi_{\mathrm{phot}}(x, t)|^2$ for resonant-wavevector excitation to the UPB. Red lines indicate the expectation value of the position operator. **c** MSDs extracted from polaritonic real-space resolved densities in (**a**) and (**b**). **d** Momentum-resolved cavity mode populations corresponding to EP propagations at $\kappa = 80$ meV in (**a**) and (**b**). Yellow lines indicate the expectation value of the cavity-mode momentum operator.

(Fig. 5f, lower panel) substantiate this, since enhancement is strongly linked to the photonic subsystem, whereas the molecular part propagates more slowly.

We note that none of these features are an artifact of averaging over only a small number of disorder realizations as all individual realizations display the same qualitative behavior (cf. Supplementary Fig. 5). Moreover, the main transport characteristics arising from the co-presence of vibronic coupling ("dynamic disorder") and inhomogeneous broadening ("static disorder") are closely related to EP transport in the presence of dynamical stochastic noise[51]. Both systems exhibit a ballistic-to-diffusive transition, symmetrization and broadening of the photonic k-space distribution, and an extended ballistic phase due to noise or vibronic coupling.

## Impact of radiative decay

Finally, we introduce cavity losses to the cavity-molecule system since current FP cavity setups based on high-reflectivity metallic mirrors have radiative lifetimes of typically 10 fs to 30 fs due to cavity photon leakage to the electromagnetic continuum. This fundamentally limits the lifetime of polaritonic state, and thus has substantial impact on the transport properties of cavity-molecule systems. To single out the impact of radiative decay and its interplay with vibronic interactions, we employ the ideal crystal model without inhomogeneous broadening in the following.

We include spontaneous cavity losses in ML-MCTDH propagations through non-Hermitian damping terms $-\sum_{k_x} i \frac{\Gamma_{k_x}}{2} \hat{a}_{k_x}^\dagger \hat{a}_{k_x}$[29,52]. A uniform decay constant $\Gamma_C = 1/\tau_C$ is assumed for all cavity modes, with apparent cavity lifetime $\tau_C = 24$ fs. As pointed out in earlier works, this ansatz is equivalent to propagating the density matrix according to a Lindblad master equation if the system stays within the electronic-photonic single-excitation subspace[53,54]. In our simulations, coupling strengths are sufficiently low and external pulses sufficiently short and weak such that this condition is met

(Supplementary Fig. 3). When the norm of the polaritonic wavepacket has decreased below 1.5% of the initial laser-excited population, the wavepacket is considered to have decayed and MSDs are not computed beyond this point.

We compare the spatiotemporal evolution of five coherently excited LP wavepackets with increasing exciton content, tuning from the photonic (pulse 1) to the excitonic regime (pulse 6). While high photonic character of the initially excited polaritonic wavepacket protects transport against the impact of vibronic coupling, it makes EP transport highly vulnerable towards radiative decay in low-Q cavities. Since the emission rate is proportional to the photon number, the polaritonic density rapidly decays in a lossy cavity. This limits the observable transport to less than 100 fs after the pulse maximum, as seen in Fig. 6a, b.

Moreover, our simulations in a lossy cavity indicate a significantly prolonged propagation when targeting the high in-plane momentum region of the LPB in Fig. 6b. Due to their excitonic nature, the impact of cavity losses on their lifetime is mitigated compared to the low-wavevector regime. Yet, vibronic interactions connect them to the more photonic bottleneck region of the LPB. Thus, exciton-like large-wavevector LP states can act as a reservoir from which population is fed to the fast-propagating polaritonic manifold. This leads to long-lived and efficient transport due to−not despite of−vibronic coupling. Our results are in line with a recent experiment[12], which could support the exciton reservoir hypothesis[26,55] as well.

In a lossy cavity, the interplay of vibronic effects with the varying group velocities and lifetimes along the EP dispersion curve creates distinct transport regimes on the LPB. The low-wavevector region (pulse 1) is characterized by low to moderate group velocities and rapid radiative decay, resulting in short-lived and short-distance polariton transport. Entering the bottleneck region (pulse 2), the group velocity has a maximum, but propagation duration is still strongly limited due to appreciable photonic content. Around the

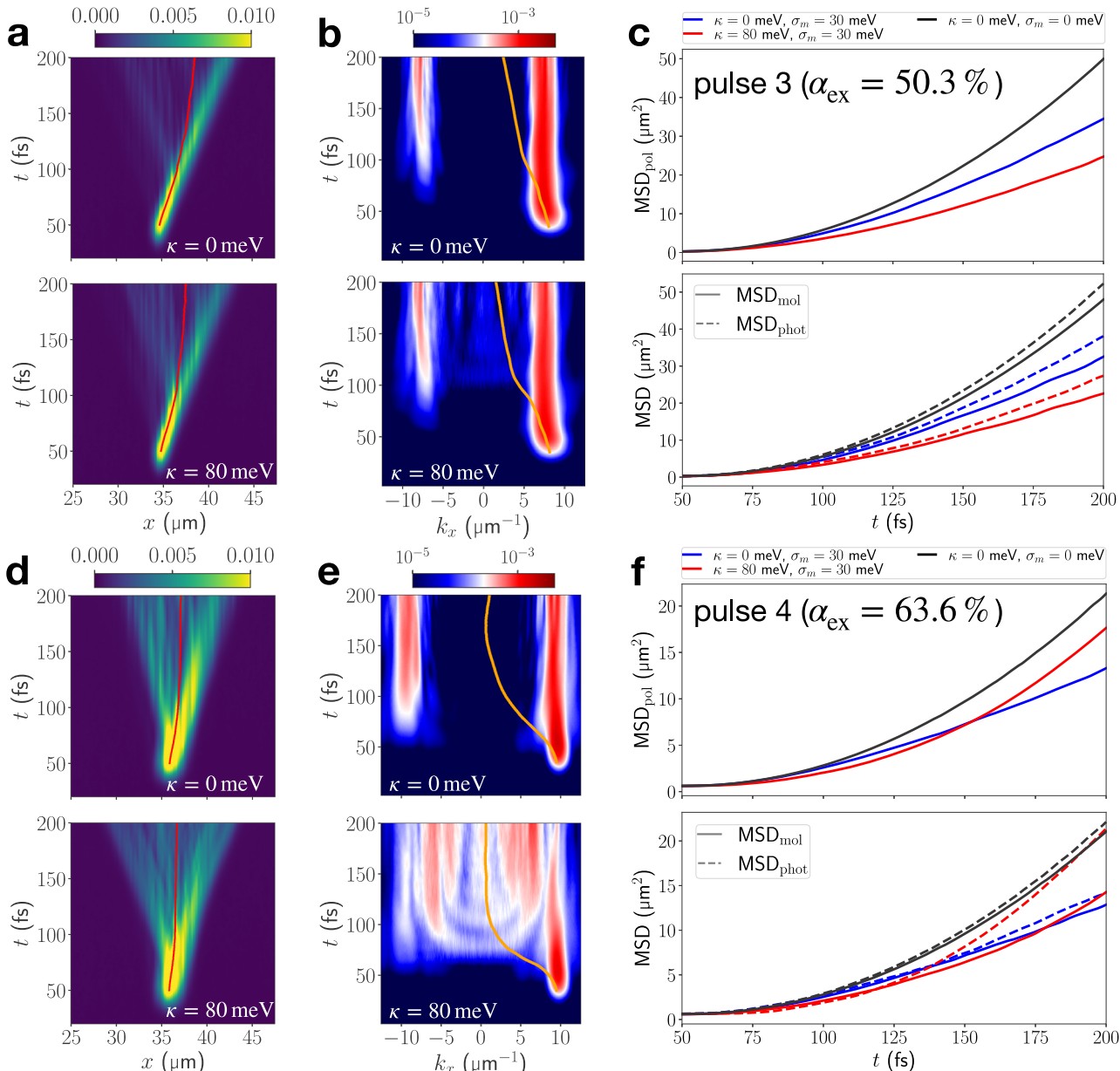

**Fig. 5 | EP propagation in the presence of static energetic disorder. a** Real-space resolved total polaritonic density $|\psi_{pol}(x, t)|^2$ for resonant-excitation to the LPB by pulse 3, at vibronic coupling strengths $\kappa = 0$ meV and $\kappa = 80$ meV and $\sigma_m = 30$ meV. Red lines indicate the expectation value of the position operator. **b** Momentum-resolved cavity mode populations corresponding to real-space densities in (**a**). Orange lines indicate the expectation value of the cavity-mode momentum resonant wavevector (pulse 3), group velocities are still high and the operator. **c** MSDs extracted from polaritonic as well as molecular and photonic real-space resolved densities for resonant-excitation by pulse 3. **d**–**f** same as (**a**–**c**) but targeting the LPB by pulse 4. For pulse 4 an increased spatial FHWM ($F_x = 1125$ nm) is used to ensure that the resulting excitation in k-space stays within the momentum-range of the bidirectional cavity model.

resonant wavevector (pulse 3), group velocities are still high and the increased excitonic content protects polaritons from cavity losses, enabling long-distance transport. Our simulations indicate that vibronic coupling has only minor impact on polariton motion in the regimes left of the resonant wavevector.

Remarkably, transport properties of lower-band polaritons to the right of the resonant wavevector (pulse 5) are improved in the presence of vibronic coupling by combining their long lifetimes and efficient access to the high-mobility bottleneck region of the LPB. This is illustrated by comparing polariton transport ensuing pulse 1 and pulse 5 (Fig. 6). Although both pulses target LPB states with very similar group velocities, the initially exciton-like wavepacket (pulse 5) propagates much longer and further than the quickly decaying photon-like wavepacket (pulse 1). When targeting even higher wavevectors, the

vanishing photonic contribution and small group velocity begin to dominate (pulse 6). Overall, this affords an optimal transport regime on the LPB between exciton contents of approximately 50% and 80%. Initial photoexcitation to this region harnesses beneficial vibronic effects and favorable transport properties set by the polariton dispersion relation. As seen in Fig. 6b, LPB wavepackets with ~50% (pulse 3) and ~80% (pulse 5) exciton content display the longest propagation distances after 200 fs.

## Discussion

We have simulated exciton-polariton transport in a coupled multi-mode cavity, treating molecular vibrations, electronic states and cavity photons as quantum degrees of freedom. In this work, fully-quantum dynamical simulations of EP transport are made possible by

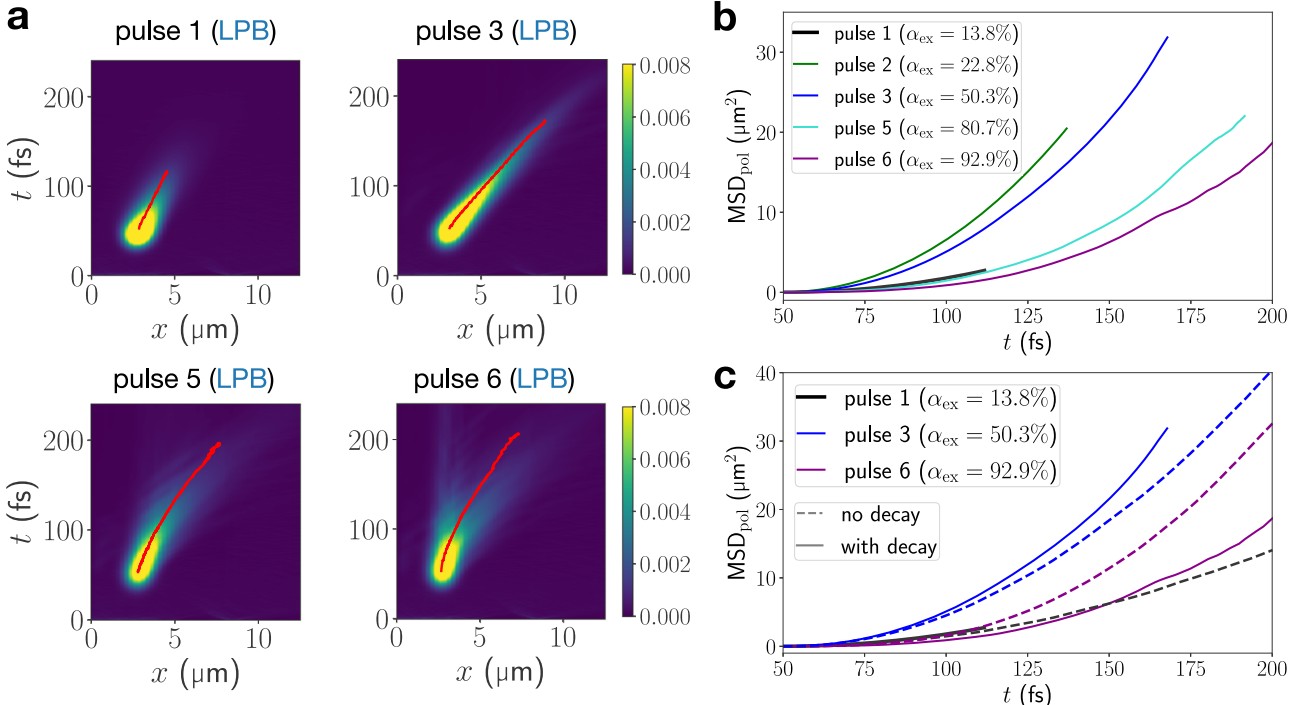

**Fig. 6 | Impact of radiative decay on polariton transport. a** Real-space resolved polaritonic densities $|\psi_{pol}(x, t)|^2$ for $\kappa = 80$ meV and three resonantly excited LPB wavepackets in a lossy cavity with $\tau_C = 24$ fs. **b** MSD of propagating EP for a vibronic coupling strength of $\kappa = 80$ meV. EPs have been excited resonantly to five different points at the LPB by pulses 1–3,5,6; exciton contents $\alpha_{ex}$ at these targeted points are given in the legend. Red lines indicate the expectation value of the position operator. **c** Comparison of MSDs for pulses 1,3,6 in the presence and absence of radiative decay.

state-of-the art quantum-dynamical methods based on a compact tree-tensor network wavefunction ansatz.

Our results show a strong dependence of polariton transport dynamics on vibronic interactions within the molecules: intramolecular vibrations mediate both inter- and intraband relaxation dynamics whose impact on EP transport depends strongly on the photon energy and wavevector of the incoming laser light. When targeting exciton-like polaritons, vibronic coupling leads to efficient population transfer toward fast-propagating polaritons at the bottleneck of the LPB. Static disorder results in localization and backscattering of the propagating polaritonic wave, with a fast transition to the diffusive regime. Crucially, when the initial excitation targets the excitonic tail at large in-plane momentum in the LPB, vibronic coupling contributes to overcome the early transition to the diffusive regime by coupling to faster propagating states with higher photonic content in the bottleneck region.

In lossy cavities, vibronic coupling prolongs the lifetime of bottleneck-polaritons since the bottleneck region is populated transiently from long-lived excitonic states. This enables long-range polariton transport on the LPB even in the presence of cavity photon decay. Thereby, our simulations highlight the major importance of molecular photophysical properties for efficient EP transport.

Previously, multi-scale semi-classical dynamics simulations[17] found enhanced mobility at 300 K when comparing with simulations at 0 K (frozen nuclei), both under off-resonant pump conditions. This highlighted the role of vibrations in transferring population from stationary bare excitonic states to propagating bright polaritonic states. In these semi-classical simulations, ballistic propagation at short times is always superseded by the onset of diffusive transport after about 100 fs (cf. Fig. 3 in ref. 17).

Our quantum dynamics simulations indicate ballistic transport for all vibronic coupling strengths when exciting the LPB in the absence of static disorder. Vibronic coupling provides mobility to the wavepacket in k-space, whereas cavity losses decrease the population of highly

photonic components, thus effectively decreasing the propagation velocity (cf. Fig. 6c), in agreement with the simulations in ref. 17. We observed the onset of diffusive transport only when resonantly exciting the UPB. In this case, the wavepacket quickly decays into the dark states manifold due to vibronic coupling. This leads to an apparent polariton wavepacket contraction in the presence of cavity losses (Supplementary Fig. 4), confirming the analysis based on semi-classical simulations in ref. 29: the stationary dark-state population persists while the propagating fraction of the wavepacket decays quickly due to its higher photonic content.

When static disorder is included in our quantum dynamics simulations, we also observe the diffusive transport regime on the LPB. For excitation near the resonant wavevector, quantum dynamical and semiclassical simulations are in good agreement (cf. Fig. 3c in ref. 17 and Fig. 5c). Both simulations indicate a transition from ballistic to diffusive transport after approximately 100 fs, along with pronounced features of localization in real-space densities.

Experimental findings on EP transport vary strongly[12,14,16,20] suggesting preserved ballistic flow or slower-than-expected diffusive transport, while a unified explanation is still lacking. These ambiguities point at a more complex mechanism behind exciton-polariton propagation which demands a clear distinction between individual effects. More precisely, dynamic disorder due to vibronic coupling, and due to thermal motion of the nuclei in the ground state, as well as static disorder due to inhomogeneities in the solid phase can have distinct individual and cooperative effects on transport. Vibronic, thermal, and static disorder effects can be cleanly separated in a fully-quantum description, but become intermingled in semi-classical approaches.

Our rigorous treatment of vibronic coupling indicates that electron-phonon scattering due to intramolecular vibronic coupling alone cannot fully account for velocity reduction and the transition from ballistic to diffusive transport. Recent experiments have already pointed out at the importance of thermal effects in EP transport. At very low temperature (5 K) the transport remains ballistic, as in our

simulations at 0 K. The onset of diffusive transport is observed at room temperature[12].

Thus, we speculate that vibronic effects are responsible for enabling the mobility of the EP wavepacket in k-space, where it accumulates in the bottleneck region, whereas thermal and static disorder play a critical role in reducing propagation velocity and limiting the ballistic flow observed in experiments.

Nonetheless, the number of simulated molecules within the quantization length is smaller than in experiments[55] and in previous semi-classical calculations[17]. This might underestimate the impact of dark states in our simulations. Participation of the dark-state manifold may be thermally activated as suggested in refs. 13,56, and can result in the onset of diffusive transport[17]. The exact mechanism for the emergence of diffusive transport at moderate to high temperatures is still a matter of debate.

The insights from this work pave the way toward a detailed mechanistic picture of polariton transport and relaxation in increasingly complex systems from first-principles. Future work in our laboratory will focus on the role of finite temperature, and the impact of dark states. Developing a microscopic understanding of polariton propagation is crucial for identifying materials with properties suited for efficient and functional polaritonic devices.

## Methods

### Organic microcavity model
In the following, we give a brief overview over the theory of EPs and the details of our organic microcavity model. We consider a one-dimensional array of $N$ identical molecules inside the polarization-plane of FP cavity, described by the Hamiltonian

$$\hat{H} = \hat{H}_{\text{cav}} + \hat{H}_{\text{mol}} + \hat{H}_{\text{cav-mol}}. \tag{1}$$

The electromagnetic field is assumed to be confined in the $z$-direction, and molecules are placed equidistantly along the $x$-axis as depicted in Fig. 1a. Periodic boundary conditions along $x$ are assumed. Following recent molecular dynamics studies on polariton transport[17,29], we further restrict the FP cavity model to a single polarization direction, $\vec{\epsilon}_y$ and a single propagation direction ($k_x \geq 0$). Thus, the Hamiltonian of the cavity field reads

$$\hat{H}_{\text{cav}} = \sum_{k_x} \omega_{\text{cav}}(k_x) \hat{a}_{k_x}^\dagger \hat{a}_{k_x} \tag{2}$$

with bosonic creation (annihilation) operators $\hat{a}_{k_x}^\dagger$ ($\hat{a}_{k_x}$) which create (annihilate) a cavity photon with in-plane momentum $k_x$ and energy $\omega_{\text{cav}}(k_x)$, given by the cavity dispersion relation $\omega_{\text{cav}}(k_x) = \sqrt{\omega_0^2 + c^2 k_x^2/n^2}$, where $\hbar\omega_0$ is the cavity mode energy at normal incidence ($k_x = 0$), $c$ the speed of light and $n$ the refractive index of the medium, which is taken as $n = 1$ for simplicity. Importantly, an appropriate discretization and truncation of in-plane momentum values $k_x$ is required in practice. Note that imposing periodic boundary conditions naturally discretizes in-plane momenta according to $k_{x,p} = 2\pi p/L_x$ where $L_x = N\Delta x$ and $p = 0, 1, ..., M-1$ [57]. In the Section "Impact of static disorder" the restriction $k_x \geq 0$ is removed, and a bidirectional cavity model with $p = -\frac{M}{2}, -\frac{M}{2}+1, ..., \frac{M}{2}-1$ is employed. In all simulations, we account for $N = 256$ molecules and include $M = 256$ photonic modes. Convergence with respect to these numbers has been checked in advance (Supplementary Figs. 1 and 2).

Each molecule is described by a Holstein-type model Hamiltonian consisting of electronic ground ($S_0$) and excited state ($S_1$) as well as one vibrational degree of freedom $Q_j$. For both electronic states, harmonic potential energy curves (PEC) along $Q_j$ with identical vibrational frequency $\omega_{\text{vib}}$ are employed. Varying horizontal displacements of the

excited state PEC with respect to the ground state PEC are investigated, which enter the molecular Hamiltonian as linear vibronic coupling constants $\kappa$. Consequently, the molecular Hamiltonian reads

$$\hat{H}_{\text{mol}} = \sum_{j=0}^{N-1} \omega_m \hat{\sigma}_j^\dagger \hat{\sigma}_j + \frac{\omega_{\text{vib}}}{2} \left( -\frac{\partial^2}{\partial Q_j^2} + Q_j^2 \right) + \kappa Q_j \hat{\sigma}_j^\dagger \hat{\sigma}_j, \tag{3}$$

where $\hat{\sigma}_j^\dagger$ ($\hat{\sigma}_j$) create (annihilate) an electronic excitation (exciton) at the $j$-th molecule in the $S_1$ state with vertical electronic excitation energy $\omega_m$ at the Franck-Condon point. The vibronic coupling constants $\kappa$ considered here (0, 50, 80 meV) correspond to Huang-Rhys factors of $S = 0.0, 0.15, 0.58$, respectively, reflecting the vibronic couplings in typical organic molecules such as pyrazine, pyrene, or Rhodamine derivatives[58–60]. Furthermore, since a sufficiently large distance $\Delta x$ is assumed, intermolecular interactions (for instance, exciton-exciton coupling) can be neglected. This prohibits any intermolecular exciton transport in the absence of coupling to the cavity.

The interaction between the cavity field and molecular excitation is described without invoking the rotating wave approximation[17,61] by

$$\hat{H}_{\text{cav-mol}} = \sum_{j=0}^{N-1} \sum_{k_x} g_j(k_x) \left( \hat{a}_{k_x}^\dagger e^{-ik_x x_j} + \hat{a}_{k_x} e^{ik_x x_j} \right) \left( \hat{\sigma}_j^\dagger + \hat{\sigma}_j \right). \tag{4}$$

Here, the coupling strength of the $j$-th molecule at position $x_j$ to the cavity mode with in-plane momentum $k_x$ is given through

$$g_j(k_x) = -\mu_{01} \cos(\theta_j) \sqrt{\frac{\omega_{\text{cav}}(k_x)}{2\epsilon_0 V_{\text{cav}}}}, \tag{5}$$

with molecular electronic transition dipole moment (TDM) at FC point $\mu_{01}$, cavity volume $V_{\text{cav}}$, and angle between the TDM vector and cavity polarization direction $\theta_j$. Cavity and molecular parameters (Table 1) are chosen to closely resemble the Rhodamine microcavity model of ref. 17, albeit with a simpler vibrational structure.

We consider a model of an ideal organic crystal without static disorder as well as a model of a disordered organic material. In the ideal crystal case, vibrational frequency, vibronic coupling strength, and electronic excitation energy are identical for all molecules, and all TDMs are aligned with the cavity mode polarization direction. To model a disordered organic material, static energetic disorder is included by sampling molecular electronic excitation energies, $\omega_m^j$, from a Gaussian distribution with standard deviation $\sigma_m$ and mean value $\bar{\omega}_m$.

Within the combined electronic-photonic single-excitation subspace, the Hamiltonian $\hat{H}$ can be diagonalized analytically to give the dispersion relation of upper and lower polaritonic branches (UPB and LPB) $\omega(k_x)$, as well as the Hopfield coefficients $|\alpha_{\text{LP/UP}}(k_x)|^2$ and $|\beta_{\text{LP/UP}}(k_x)|^2$, i.e., the contributions of cavity modes and molecular excitons to the lower polaritonic and upper polaritonic (LP and UP) states, respectively. From the dispersion relation, the in-plane group velocity can be obtained via $v_{\text{gr}}^{\text{LP/UP}} = \partial\omega_{\text{LP/UP}}/\partial k_x$.

### Quantum dynamics simulation
The details of our quantum-dynamical approach, including the ML-MCTDH tree structure for the cavity-molecule wavefunction, and the treatment of non-local couplings between delocalized cavity modes and localized molecular excitations in Eq. (4) will be discussed in a future publication. In short, transforming the cavity mode basis from momentum to real-space is beneficial as it results in localized couplings and dramatically reduces the number of Hamiltonian terms. The momentum and real-space bases are related through a unitary discrete Fourier transformation, through which one can change between momentum and real-space representations of the cavity wavefunction,

$\psi_{\text{phot}}(k_{x,p}, t)$ and $\psi_{\text{phot}}(x_j, t)$, respectively. A similar transformation has been employed in a recent publication[62] to simulate polaritons beyond the long-wavelength approximation.

In the section "Resonant Excitation", an external laser pulse explicitly incorporated through a semiclassical dipolar laser-molecule interaction Hamiltonian, $\hat{H}_{\text{las}} = -\sum_j \mathcal{E}(x_j, t)(e^{ik_x^{(0)}x_j}\hat{\sigma}_j^\dagger + e^{-ik_x^{(0)}x_j}\hat{\sigma}_j)$. Temporal and spatial Gaussian envelope functions, $A(t)$ and $B(x_j)$, for the laser field $\mathcal{E}(x_j, t) = A(t)B(x_j)\cos(\omega_L t)$ result in a finite width in energy and momentum space. Laser intensities are chosen low enough to stay within the linear absorption regime, i.e., only one photon is absorbed (cf. Supplementary Fig. 3). All relevant model and laser parameters are summarized in Table 1.

## Data availability
The raw data that support the findings of this study are available in Zenodo with the identifier https://doi.org/10.5281/zenodo.15235668.

## Code availability
The MCTDH code with its full documentation and any further input files needed to reproduce particular results of the current contribution are available upon request from the authors.

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

## Acknowledgements

The authors acknowledge support by the state of Baden-Württemberg through bwHPC and the German Research Foundation (DFG) through grant INST 35/1597-1 FUGG. O.V. acknowledges support through the German Research Foundation (DFG) Collaborative Research Center 1249.

## Author contributions

N.K., G.G. and O.V. conceived the idea and planned the calculations and analysis. N.K. performed and analyzed the calculations and numerical simulations and composed the first version of the manuscript. All authors contributed to the interpretation of the results and to the final version of the manuscript.

## Funding

## Competing interests

The authors declare no competing interests.
