## [Transparent Peer Review file · Nature Communications]

Quantum dynamics simulation of exciton-polariton transport

Corresponding Author: Professor Oriol Vendrell

Version 0:

Reviewer comments:

Reviewer #1

(Remarks to the Author)

In this manuscript, the authors perform an exact quantum dynamical simulation of exciton polaritons coupled to molecular vibrations at $T = 0\text{K}$. Recent experiments and other theoretical work (including many by the authors themselves) has revealed that exciton polariton transport can be ballistic, diffusive, or have renormalized group velocities. Most of these effects now have some theoretical understanding, at least at the semi-classical/mixed quantum classical level.

What sets this paper apart from previous work is that this most likely, for the first time, employed an exact quantum dynamical approach to simulate phonon-exciton-polariton dynamics. However, they also had to make approximations in constructing the model, i.e., dilute a sample of molecules (the distance between molecules is very high) and set the temperature to 0, which do limit the scope of the insights obtained here.

Overall, the work is interesting, and the numerical aspect is commendable. However, at this point I cannot recommend the publication of this work as I have a few concerns.

1. The authors mention the ballistic to diffusion transition as a key feature of the recent experiments that the authors cite. However, at $T = 0\text{K}$, I expect even excitonic materials to exhibit ballistic motion. In fact, previous experimental works observe diffusive motion (<https://www.nature.com/articles/s41563-022-01463-3>) even when resonantly exciting to the LPB. Here, on the other hand, the authors find only ballistic motion. At the same time, previous works also find a group velocity renormalization (<https://www.nature.com/articles/s41467-023-39550-x>) when exciting to the LPB. I would be curious to know what the authors observe in terms of velocity renormalization. Does such an effect still exist even at 0K ?

2. As the authors mention, they use few molecules (250ish) in their simulations at a very large distance. The light-matter couplings are proportional to $\sqrt{N/V}$, which means the rabi-splitting is directly proportional to the concentration of the molecules. This would mean that the authors are scaling individual molecular coupling by at least ~ 100 times than they are in an experimental setup. The authors should discuss the validity of their result under such approximations.

3. The authors mention the real-space description of cavity modes. The authors should cite recent work that utilized such description : <https://arxiv.org/pdf/2409.07992>.

Reviewer #2

(Remarks to the Author)

The manuscript explores the numerical investigation of transport processes under strong exciton-photon coupling, with a particular focus on the effects of vibronic coupling on transport dynamics. Enhanced exciton transport is a highly active and rapidly evolving research area, with significant efforts aimed at fully understanding the various factors influencing transport in exciton-polariton systems. The authors employ fully quantum dynamical simulations that account for both the vibrational degrees of freedom of the molecules and the multimode, lossy nature of the optical cavity resonance. They conduct a

thorough study under different vibronic coupling strengths and excitation conditions (both resonant and non-resonant), identifying the momentum-relaxation pathways as well as scenarios where vibronic coupling can either enhance or hinder polariton transport.

Overall, the results presented in this manuscript are convincing and align rather well with experimental observations, marking an important step towards realistic modeling of polaritonic transport in complex organic cavities—a highly sought-after goal in the field. While the work is interesting and the quantum mechanical treatment of molecular vibrations is certainly appreciated, the novelty of the results may not meet the high standards required for publication in Nature Communications. Many features observed in the simulations, such as intra- and inter-band population relaxation and the bottleneck effect, have been captured by earlier theoretical studies cited by the authors. Additionally, the model's representation of a concrete physical system is unclear. The current model does not include static disorder, which is significant in common organic systems with simple dye molecules (e.g., Rhodamine, as mentioned in the text), nor does it account for direct coupling between adjacent sites, which is necessary for modeling transport in crystalline materials. The authors indicate plans to include these effects in future studies. Still, without them, the scope of the work may be too narrow to constitute a major advancement justifying publication in Nature Communications.

Apart from these general comments, there are a few, more technical points that the authors may wish to address:

1. In p. 1 the authors mention references 12, 15 and 20 as examples where only ballistic flow was observed, noting that these experimental studies were conducted under low (cryogenic) temperatures. While this is not wrong, it is also worthwhile to remember that these studies involved crystalline materials (inorganic semiconductors and halide perovskites). For common organic molecules, it is most probable that static disorder would lead to velocity reduction and diffusion even in the low-temperature regime.

2. With an intermolecular distance Δ_x of 250 nm and using periodic boundary conditions, I would expect that the maximal wavenumber supported by the grid (following the Nyquist theorem) would be $\pi/\Delta_x = 12.56 \mu\text{m}^{-1}$, with the discrete set of 256 in-plane momenta spanning both negative and positive values. However, in the graphs k_x extends from 0 to $25 \mu\text{m}^{-1}$. Am I missing something here?

3. In p. 4, right column, the authors refer to “time-dependent mean-position $\langle x_{\text{pol}} \rangle(t)$ in Fig. 1a”. I believe it is meant to be Fig. 2a.

4. For the dynamics shown in Fig. 4b, the authors claim that after UP excitation the wavepacket experiences a slowing down and diffusive transport, based on the evolution of the mean position and MSD. However, looking at the time-dependent spatial distribution it seems that the wavepacket simply splits into two lobes, where one remains almost stationary while the other moves at a constant velocity. In fact, the evolution is not very different than what is observed in Fig. 3a for pulse 5 or Fig. 3b for pulse 6. Could the seemingly diffusive behavior be an artifact of treating the wavepacket as a single distribution? Similarly, the behavior seen in Fig. 4a, bottom panel, also seems more like a main lobe moving at a constant velocity while leaving a weak, stretched tail behind it.

5. The authors may consider discussing relevant results from the Cao group, published in 10.1103/PhysRevLett.130.213602 and 10.48550/arXiv.2410.11051.

Reviewer #3

(Remarks to the Author)

Reviewer #4

(Remarks to the Author)

This manuscript, Niclas Krupp et al “Quantum dynamics simulation of exciton-polariton transport”, investigates the roles of vibronic interactions during the ballistic transport of exciton-polaritons. The authors construct a fully quantum model with one dispersion band of cavity photon modes (with built-in group velocity), one set of molecular modes with intramolecular vibronic coupling, and the cavity-molecule coupling terms that depend on angles. They propagate the wavefunction with the ML-MCTDH method and evaluate the expectation value of the position and momentum of the molecular component and cavity component respectively. They prepare initial conditions as selectively pumping varying regions on both upper and lower polariton branches and find that when the intramolecular coupling is turned on, (i) the spatial propagation of polaritonic density is drastically altered (and in some cases accelerated), (ii) the momentum of cavity mode drifts for cases with large photonic component (Hopfield coefficient). Overall the paper is well-written. I am certain that the manuscript will become suitable for publication after my questions and concerns are swiftly addressed.

Please further clarify the two concepts: photonic component $|\psi_{\text{phot}}(x_j, t)|^2$ and $|\psi_{\text{phot}}(k(x, p), t)|^2$ and their relations. Are they the same entity with different representations?

Could the author please explain the choice of refractive index $n = 1.0$ in Table 1, as the cavity is filled with molecules strongly interacting with light? Perhaps the n in the cavity dispersion relation can be removed from the formula and assume c to be smaller than the vacuum speed of light.

The finding of strong acceleration of polariton transport is really interesting, which arises from the wavevector shift due to relaxation induced by intramolecular vibronic interaction. However, I am uncertain about my understanding of the underlying physics here. If we compare the results of pulse 5 and 6, the first result looks reasonable as the polariton is dissipating its energy and momentum to the molecular vibrational mode; the second, however, seems strange as the majority of the population moves to higher momentum regions. This seems to be contradictory to the standard polariton condensate picture as the energy and momentum should mostly dissipate to the lowest LP region. Is this a manifestation of a small and low-temperature model?

I am curious about the dominant entity in Fig 3d lower left that possesses $k_x \approx 5-10 \mu\text{m}^{-1}$. Does the entity (that travels fast) correspond to point 7 in the upper polariton branch or point 3 in the lower polariton branch in Fig 1b?

Moreover, following the previous question, I presume the method conserves the overall momentum, what is the source of this momentum shift (unidirectionally losing or gaining about $5 \mu\text{m}^{-1}$) to reach the bottleneck regime)? This is not obvious to me, especially the cavity mode only has positive k_x , if I want to think of polariton-polariton scattering. Perhaps this is irrelevant. But please explain a bit about the uniqueness of this wavevector range $k_x \approx 5-10 \mu\text{m}^{-1}$. Why don't I see a similar effect for pulse 8?

Following question 3, is the temperature still low? I believe a certain amount of energy (~ 200 meV) is quenched into the molecular vibrational mode. I do not have a very good intuition but it would be very informative if the authors can quickly extract the evolution of effective temperature in one or several representative molecules vibrational mode (Q_j). (If the authors find it not interesting or insightful, perhaps the figure and/or discussion can be appended in the supplementary information.)

The supplementary information needs better documentation. There seem to be only figures captions and I would very much appreciate it if the authors could append more discussions of the figures in the supplementary information.

Version 1:

Reviewer comments:

Reviewer #1

(Remarks to the Author)

The replies provided by the authors address all my concerns. Together with the changes they made to the main text and supplementary information, I recommend the publication of the current version of the manuscript in Nature Communications.

One extra comment to authors: it seems there is some issue with the compilation of the manuscript; after page 12, formatting is a bit broken.

Reviewer #2

(Remarks to the Author)

The authors have adequately addressed all the reviewers' comments and questions and improved their manuscript by extending its scope and providing additional simulation results and clarifications. I find that the revised manuscript is suitable for publication in Nature Communications.

Reviewer #3

(Remarks to the Author)

Reviewer #4

(Remarks to the Author)

I would like to thank the authors for the detailed response. Although some answers are not satisfactory, I think that overall speaking, the manuscript is ready to publish.

Reply letter:
Quantum dynamics simulation of exciton-polariton transport
Niclas Krupp, Gerrit Groenhof, Oriol Vendrell

Reviewer #1 (Remarks to the Author)

In this manuscript, the authors perform an exact quantum dynamical simulation of exciton polaritons coupled to molecular vibrations at $T = 0$ K. Recent experiments and other theoretical work (including many by the authors themselves) has revealed that exciton polariton transport can be ballistic, diffusive, or have renormalized group velocities. Most of these effects now have some theoretical understanding, at least at the semi-classical/mixed quantum classical level.

What sets this paper apart from previous work is that this most likely, for the first time, employed an exact quantum dynamical approach to simulate phonon-exciton-polariton dynamics. However, they also had to make approximations in constructing the model, i.e., dilute a sample of molecules (the distance between molecules is very high) and set the temperature to 0, which do limit the scope of the insights obtained here.

Overall, the work is interesting, and the numerical aspect is commendable. However, at this point I cannot recommend the publication of this work as I have a few concerns.

We thank the Referee for the careful assessment of our work and the useful critical remarks, which we are addressing below.

1. The authors mention the ballistic to diffusion transition as a key feature of the recent experiments that the authors cite. However, at $T = 0$ K, I expect even excitonic materials to exhibit ballistic motion. In fact, previous experimental works observe diffusive motion (<https://www.nature.com/articles/s41563-022-01463-3>) even when resonantly exciting to the LPB. Here, on the other hand, the authors find only ballistic motion.

The Referee points out to observation of diffusive motion in highly excitonic states of a photonic crystal system. As the Referee points out below, at $T = 0$ K ballistic motion without renormalization has been observed for other kinds of systems, for example a perovskite-based material in the reference mentioned in the next paragraph.

While the reviewer may be correct that at 0 K also exciton transport could be ballistic (as the weak excitonic coupling would exceed the thermal energy) we do not understand how this connects to the paper by Balasubrahmaniyam et al., as the latter describes polariton transport at room temperature. Under such conditions, they observe both ballistic and diffusive transport for TDBC J-aggregates coupled to a Bloch Surface Wave, with a transition from ballistic propagation to diffusion, when the BSW contribution to the polariton states decreases. The ballistic motion observed in our simulations are thus in line with these experiments, as well as with the experiments performed under cryogenic conditions on a perovskite coupled to a Fabry-Pérot micro-cavity by Xu et al.

At the same time, previous works also find a group velocity renormalization (<https://www.nature.com/articles/s41467-023-39550-x>) when exciting to the LPB. I would be curious to know what the authors observe in terms of velocity renormalization. Does such an effect still exist even at 0K ?

The reference mentioned above indicates almost no renormalization (i.e. slow down still within the ballistic regime) of the transport velocity at the cryogenic temperature of 5 K. This is in good agreement with our simulations, which also indicate little change in propagation velocity compared to the slope of the dispersion relation at 0 K.

The reviewer suggests that in the paper of Xu, velocity renormalization was observed for a strongly coupled perovskite material when the LPB is resonantly excited. Indeed, at room temperature, such renormalization occurs, and was attributed to polariton-phonon (vibration) interactions. However, in the same article, the authors also share results of microscopy measurements 5 K, which matches more closely the conditions of our simulations, and show that the measured velocities match the group velocities within experimental error. These observations are in good agreement with the results of our simulations, which suggest that under cryogenic conditions, the transport is ballistic. Based on previous measurements of Xu et al., and our simulation results, we conclude that the re-normalization effect does

not play a role at temperatures below 5 K.

2. As the authors mention, they use few molecules (250ish) in their simulations at a very large distance. The light-matter couplings are proportional to $\sqrt{(N/V)}$, which means the Rabi-splitting is directly proportional to the concentration of the molecules. This would mean that the authors are scaling individual molecular coupling by at least ~ 100 times than they are in an experimental setup. The authors should discuss the validity of their result under such approximations.

The Referee is correct in that the coupling per molecule is substantially higher than the actual coupling at the molecular density of typical experiments. This is the case for most simulations of polaritonic systems, where the number of molecules can be seen as a convergence parameter of the simulation if the single-molecule coupling is scaled by $1/\sqrt{N}$ as to maintain a constant collective coupling.

Considering the simulations in the reference mentioned by the Referee, there the authors ran simulations with 6001 molecules. Therefore, taking the reviewer's estimate of a 100-fold higher field than in reality, that would imply that there would be 25,000,000 molecules coupled in a real cavity. In that sense, while 6001 is more than 256, the scaling of the cavity vacuum field strengths in the mentioned work would be a factor of 20. Less than 100, but roughly within the same order of magnitude.

The Referee is nonetheless asking us to discuss the validity of such approximation. We point the Referee to the first section of the supporting material, where we have performed a convergence analysis of the mean position of the photonic and excitonic excitations, while keeping the intermolecular distance constant to $\Delta x = 250$ nm and correspondingly the quantization length of the periodic box being $L = (N - 1)\Delta x$. In the analysis, we show that convergence is achieved for about 128 molecules, and the difference between 256 and 512 is insignificant. Hence, for this intermolecular distance, our transport simulations in the 1-D wire are converged with respect to the number of molecules and photon modes.

It remains to be considered, what is the effect of the molecular separation for a fixed quantization length L , in closer connection with the reviewer's question. The molecular separation determines the maximum $|k_x|$ of the cavity modes, whose nodal structure can be "seen", or sampled, by the molecular lattice. For $\Delta x = 250$ nm this is roughly $25 \mu\text{m}^{-1}$ (cf. Q.2. Referee #2 below).

In the SI we describe transport simulations starting from a localized molecular excitation, where the number of molecules has been doubled from 256 to 512 while, this time, keeping L constant, i.e. increasing density. This results in a new set of molecular states which do not directly interact with the light modes because of their highly oscillatory nature. The propagation of the photonic component and the front of the molecular component remain unchanged in this case, while only the molecular subsystem has a larger non-propagating component trapped in the new states that do not see the photonic modes. The number of photonic modes can now in turn be increased to 512. In doing so, a small change of mean propagation velocity is found in the molecular component, but the change is tiny since the new molecular states and photonic modes are very much out of resonance.

Hence, we conclude that our simulations capture the key features of the polaritonic transport dynamics (at low temperature), whereas a larger number of molecules may result in a larger non-propagating component in the material subsystem and hence a faster onset of excitonic-dominated transport.

3. The authors mention the real-space description of cavity modes. The authors should cite recent work that utilized such description : <https://arxiv.org/pdf/2409.07992>.

We thank the Referee for pointing us to this reference. It describes a discrete FT of the electromagnetic modes similar to the one we used, resulting in a more sparse structure of the light-matter coupling. This reference is now cited.

Reviewer #2 (Remarks to the Author)

The manuscript explores the numerical investigation of transport processes under strong exciton-photon coupling, with a particular focus on the effects of vibronic coupling on transport dynamics. Enhanced exciton transport is a highly active and rapidly evolving research area, with significant efforts aimed at fully understanding the various factors influencing transport in exciton-polariton systems. The authors employ fully quantum dynamical simulations that account for both the vibrational degrees of freedom of the molecules and the multimode, lossy nature of the optical cavity resonance. They conduct a thorough study under different vibronic coupling strengths and excitation conditions (both resonant and non-resonant), identifying the momentum-relaxation pathways as well as scenarios where vibronic coupling can either enhance or hinder polariton transport. Overall, the results presented in this manuscript

are convincing and align rather well with experimental observations, marking an important step towards realistic modeling of polaritonic transport in complex organic cavities — a highly sought-after goal in the field. While the work is interesting and the quantum mechanical treatment of molecular vibrations is certainly appreciated, the novelty of the results may not meet the high standards required for publication in Nature Communications. Many features observed in the simulations, such as intra- and inter-band population relaxation and the bottleneck effect, have been captured by earlier theoretical studies cited by the authors. Additionally, the model's representation of a concrete physical system is unclear. The current model does not include static disorder, which is significant in common organic systems with simple dye molecules (e.g., Rhodamine, as mentioned in the text), nor does it account for direct coupling between adjacent sites, which is necessary for modeling transport in crystalline materials. The authors indicate plans to include these effects in future studies. Still, without them, the scope of the work may be too narrow to constitute a major advancement justifying publication in Nature Communications.

We thank the Referee for the detailed evaluation of our work. As the referee points out, we consider the quantized nature of the vibrational modes and include cavity losses in the picture, but had not accounted for static disorder, which is present in most organic materials.

This is an important aspect, as the referee mentions, and we have now extended the model to include simulations with static disorder, which are discussed in a new subsection.

The new simulations show how static disorder plays an important role in the transition to the diffusive regime. Importantly, disorder results in back-scattering from positive (left-to-right) to negative momenta, which was not present in the simulations without disorder. To account for the reversal in propagation direction, we have extended the model to include the negative momentum portion of the k_x -axis.

Apart from these general comments, there are a few, more technical points that the authors may wish to address:

1. In p. 1 the authors mention references 12, 15 and 20 as examples where only ballistic flow was observed, noting that these experimental studies were conducted under low (cryogenic) temperatures. While this is not wrong, it is also worthwhile to remember that these studies involved crystalline materials (inorganic semiconductors and halide perovskites). For common organic molecules, it is most probable that static disorder would lead to velocity reduction and diffusion even in the low-temperature regime.

We agree with the point made by the Referee. As mentioned above, we have added disorder to the model to more closely mimic polaritonic transport based on organic materials. In this case, transition to the diffusive regime is found in our simulations at 0 K.

2. With an intermolecular distance Δx of 250 nm and using periodic boundary conditions, I would expect that the maximal wavenumber supported by the grid (following the Nyquist theorem) would be $\pi/\Delta x = 12.56 \mu\text{m}^{-1}$, with the discrete set of 256 in-plane momenta spanning both negative and positive values. However, in the graphs k_x extends from 0 to $25 \mu\text{m}^{-1}$. Am I missing something here?

The Referee is correct in stating the relation between Δx and the maximum momentum range that this discretization supports. However, the relation between the photonic momentum and molecular spatial grid in the Hamiltonian model requires some clarification.

In the model (without disorder) we consider a set of 256 equally spaced photon momenta along k_x , ranging from 0 to $25 \mu\text{m}^{-1}$, and a set of 256 molecules placed at real-space positions between 0 and $250 \times 256 \text{ nm}$. The coupling between the delocalized mode with momentum k_x and the j -th molecule is hence proportional to $\exp(i k_x j \Delta x)$, and no use of Fourier relations has been made up to this point. In the supporting material we present simulations with an increasing number of M delocalized modes and N molecules, and show that convergence in the transport properties within the model parameters is achieved for about 128 modes for the system with 256 molecules.

Now, in order to make the simulations with vibronic coupling more efficient for 256 molecules, we consider an equal number of modes and perform a discrete transformation to real space, such that each mode's position coincides with the position of one molecule. The coupling between molecules and modes becomes sparse and quickly decreases as a function of the difference of their indices. We have numerically checked that the transformation does not introduce any artifacts.

In simulations with disorder, we need to account for back-scattering of the photon. We have therefore shifted the momentum range to span $-12.5 \mu\text{m}^{-1}$ to $12.5 \mu\text{m}^{-1}$, the range the Referee indicates above. This shift results in the same position of the localized modes and the molecules, which holds as long as one considers the same number of elements for both subspaces.

We have tested numerically (Fig. below) that no artifacts have been introduced with the new definition of the momentum coordinate, and that the previous simulations without static disorder lead numerically to the same results in the two representations of momentum.

Figure 1: Comparison between unidirectional (“uni”) and bidirectional (“bi”) cavity model. (a) Real-space resolved total polaritonic density $|\psi_{\text{pol}}(x, t)|^2$ for resonant-wavevector excitation to the LPB with pulse 3 and $\kappa = 80$ meV. (b) Corresponding time-dependent cavity mode populations $|\psi_{\text{phot}}(k_x, t)|^2$. (c)-(d) MSDs extracted from polaritonic real-space resolved densities for excitation with pulse 3 and $\kappa = 0$ meV and $\kappa = 80$ meV.

3. In p. 4, right column, the authors refer to “time-dependent mean-position $\langle x_{\text{pol}} \rangle (t)$ in Fig. 1a”. I believe it is meant to be Fig. 2a.

Thank you for spotting this. It has been corrected.

4. For the dynamics shown in Fig. 4b, the authors claim that after UP excitation the wavepacket experiences a slowing down and diffusive transport, based on the evolution of the mean position and MSD. However, looking at the time-dependent spatial distribution it seems that the wavepacket simply splits into two lobes, where one remains almost stationary while the other moves at a constant velocity. In fact, the evolution is not very different than what is observed in Fig. 3a for pulse 5 or Fig. 3b for pulse 6. Could the seemingly diffusive behavior be an artifact of treating the wavepacket as a single distribution? Similarly, the behavior seen in Fig. 4a, bottom panel, also seems more like a main lobe moving at a constant velocity while leaving a weak, stretched tail behind it.

The Referee’s point is valid. We think that there is some ambiguity in whether one describes the transport only by its MSD, or by tracking the position of the wavefront and/or the slowly propagating portion of the wavepacket, which may have separated into distinct lobes. We mention this now in the text and make clear to the Reader that the diffusive transport obtained when exciting into the UPB can be decomposed into a ballistic component where the system must have reached the LPB, and an almost static component made up by mostly excitonic population.

5. The authors may consider discussing relevant results from the Cao group, published in 10.1103/Phys-RevLett.130.213602 and 10.48550/arXiv.2410.11051.

These results are now discussed in context in the new version.

Reviewer #4 (Remarks to the Author)

This manuscript, Niclas Krupp et al “Quantum dynamics simulation of exciton-polariton transport”, investigates the roles of vibronic interactions during the ballistic transport of exciton-polaritons. The authors construct a fully quantum model with one dispersion band of cavity photon modes (with built-in group

velocity), one set of molecular modes with intramolecular vibronic coupling, and the cavity-molecule coupling terms that depend on angles. They propagate the wavefunction with the ML-MCTDH method and evaluate the expectation value of the position and momentum of the molecular component and cavity component respectively. They prepare initial conditions as selectively pumping varying regions on both upper and lower polariton branches and find that when the intramolecular coupling is turned on, (i) the spatial propagation of polaritonic density is drastically altered (and in some cases accelerated), (ii) the momentum of cavity mode drifts for cases with large photonic component (Hopfield coefficient). Overall the paper is well-written. I am certain that the manuscript will become suitable for publication after my questions and concerns are swiftly addressed.

We thank the Referee for its critical assessment of the manuscript and questions, which we are addressing in the following.

Please further clarify the two concepts: photonic component $|\psi_{phot}(x_j, t)|^2$ and $|\psi_{phot}(k(x, p), t)|^2$ and their relations. Are they the same entity with different representations?

Yes, they are different representations of the same entity. A note has been added for the Reader in this respect.

Could the author please explain the choice of refractive index $n = 1.0$ in Table 1, as the cavity is filled with molecules strongly interacting with light? Perhaps the n in the cavity dispersion relation can be removed from the formula and assume c to be smaller than the vacuum speed of light.

We take $n = 1$ for simplicity, as the simulations are not aimed at a quantitative reproduction of a specific experimental realization. A modification n in direction of 1.5, the diffraction of index of typical organic materials (doi:10.1103/PhysRevB.63.121302 e.g. give a refractive index of $n \approx 1.5-1.6$ for the organic film used in their cavity), would introduce a change in the asymptotic slope of the dispersion curve, but it would not modify the conclusions of our simulations.

The finding of strong acceleration of polariton transport is really interesting, which arises from the wavevector shift due to relaxation induced by intramolecular vibronic interaction. However, I am uncertain about my understanding of the underlying physics here. If we compare the results of pulse 5 and 6, the first result looks reasonable as the polariton is dissipating its energy and momentum to the molecular vibrational mode; the second, however, seems strange as the majority of the population moves to higher momentum regions. This seems to be contradictory to the standard polariton condensate picture as the energy and momentum should mostly dissipate to the lowest LP region. Is this a manifestation of a small and low-temperature model?

The initial states created by pulses 6 (old 5) and 7 (old 6) are in both cases highly excitonic. The initial wavepacket is initially found in a flat region of the dispersion curve and therefore moving with small group velocity. Also in both cases, the system can trade some electronic excitation energy for vibrational energy while increasing its photonic character, thus evolving into the region of the dispersion curve with the largest group velocity. This is where the density of states is largest due to the stronger coupling between the excitonic and photonic components. In both cases, the population ends up in the LP region with the largest group velocity. When starting from point 6, the system is formally already in the LP branch, whereas when starting in point 7, the system is formally in the UP branch and evolves toward the LP branch due to vibronic coupling, as Figure 3b illustrates. Hence, we believe that this picture is not contradictory with the picture that the system condensates in the LP branch, although condensation is a different process than the transport studied here.

I am curious about the dominant entity in Fig 3d lower left that possesses $k_x \approx 5 - 10 \mu m^{-1}$. Does the entity (that travels fast) correspond to point 7 in the upper polariton branch or point 3 in the lower polariton branch in Fig 1b?

It corresponds to point 7 (old 6) in the UP branch. The old point 3 was not discussed in Figure 3d. [niclas: in Figure 3d we start from the exciton-like pulse 7 (old 6) from where the system relaxes to the bottleneck on the LPB (cf. answer to previous question, and it's also described in the main text); where exactly this population is located in the picture of Fig. 1b is not so clear, somewhere between point 2 and 3]

Moreover, following the previous question, I presume the method conserves the overall momentum, what is the source of this momentum shift (unidirectionally losing or gaining about $5 \mu m^{-1}$ to reach the bottleneck regime)? This is not obvious to me, especially the cavity mode only has positive k_x , if I want to think of polariton-polariton scattering. Perhaps this is irrelevant. But please explain a bit about the uniqueness of this wavevector range $k_x \approx 5 - 10 \mu m^{-1}$. Why don't I see a similar effect for pulse 8?

We have added negative momenta and backscattering plays an important role in the presence of static disorder, as discussed in detail above. We have checked that the simulations without negative momenta and without static disorder remain identical to a high degree of accuracy with simulations including the negative momenta.

However, the question about the transfer to higher momenta and momentum conservation in the model remains a valid one. The short answer is, the simulation does not conserve the overall linear momentum of the dynamical degrees of freedom involved, but this is physically correct within the model with fixed lattice sites (note that the vibration is assumed an internal coordinate of each molecule, and hence there is no overall momentum where all particles translate through space).

A useful picture is that the photons and the excitons scatter at the molecular positions in a similar way as a ball bounces at a fixed wall and the momentum of the ball is not conserved. Total momentum is conserved for the ball and the wall (attached to the planet), but not considering the recoil of the wall is perfectly fine in practice due to its much larger mass.

Similarly, it is not contradictory that the internal excitation energy of the molecules can be exchanged for an increase in momentum k_x of the photons. If the molecules could move, this would cause a correspondingly tiny recoil of the lattice (due to the molecular mass being large).

Finally, the reviewer is also asking about (old) pulse 8; the initial state prepared by pulse 9 (old 8) is too photonic such that exchange of momentum with molecules does not happen. This has already been explained in the main text.

Following question 3, is the temperature still low? I believe a certain amount of energy (200 meV) is quenched into the molecular vibrational mode. I do not have a very good intuition but it would be very informative if the authors can quickly extract the evolution of effective temperature in one or several representative molecules vibrational mode (Q_j). (If the authors find it not interesting or insightful, perhaps the figure and/or discussion can be appended in the supplementary information.)

We have been thinking about this question, but we do not see a clear way to perform such an analysis of an effective temperature. We assume always that one photon has been absorbed, and then, as the Referee points out correctly, some hundreds of meV are transferred into vibrations. How much per vibration depends on the number of molecules in the model, how much the excitation spreads among the various molecular units, etc. Some molecules will remain completely unaffected by the excitation, whereas some others will actively participate in the transport. We do not see a clear-cut way to translate their energy into a temperature that can provide further insight.

In future studies we plan to add the effect of temperature in the quantum dynamics simulations, but this is a whole different enterprise, as the initial conditions need to be sampled with Boltzmann statistics, or use some other trick such as thermofield dynamics.

The supplementary information needs better documentation. There seem to be only figures captions and I would very much appreciate it if the authors could append more discussions of the figures in the supplementary information.

We have improved the supporting material with more detailed discussions and figures.